# Base-Mediated Claisen Rearrangement of CF_3_-Containing Bisallyl Ethers

**DOI:** 10.3390/molecules26144365

**Published:** 2021-07-19

**Authors:** Yoko Hamada, Rio Matsunaga, Tomoko Kawasaki-Takasuka, Takashi Yamazaki

**Affiliations:** Division of Applied Chemistry, Institute of Engineering, Tokyo University of Agriculture and Technology, 2-24-16 Nakamachi, Koganei 184-8588, Japan; yokoooooo30pama@gmail.com (Y.H.); rio.futa1115@gmail.com (R.M.); takasuka@cc.tuat.ac.jp (T.K.-T.)

**Keywords:** Claisen rearrangement, isomerization, trifluoromethyl, Cieplak rule

## Abstract

We have previously clarified that the strongly electron-withdrawing CF_3_ group nicely affected the base-mediated proton shift of CF_3_-containing propargylic or allylic alcohols to afford the corresponding α,β-unsaturated or saturated ketones, respectively, which was applied this time to the Claisen rearrangement after *O*-allylation of the allylic alcohols with a CF_3_ group, followed by isomerization to the corresponding allyl vinyl ethers via the proton shift, enabling the desired rearrangement in a tandem fashion, or in a stepwise manner, the latter of which was proved to have attained an excellent diastereoselectivity with the aid of a palladium catalyst.

## 1. Introduction

It is widely understood that strategic entry of fluorine atoms or fluorinated groups to adequate molecules gave strong impact to the original character in many instances, and thus the development of novel methods for the construction of a variety of such compounds has attracted significant attention of researchers working in the field of synthetic organic chemistry, material science, and biologically active compounds [1,2,3,4,5]. For this reason, we have been studying to realize facile preparation of such molecules, and recently reported an interesting proton transfer starting from both propargylic [6] and allylic alcohols [7], enabling to form diverse α,β-unsaturated and saturated ketones, respectively, just by their treatment with very convenient as well as easy-to-handle tertiary amines. The representative example for the latter was described in Scheme 1. Presence of the electron- withdrawing CF_3_ group was considered to play a crucial role in the increase of the acidity of a proton H^a^ at the allylic position of **1**. Actually its abstraction was realized by the action of such a weak base as diazabicyclo[5.4.0]undec-7-ene (DBU) under the toluene refluxing condition which resulted in the simple isomerization to the intermediates **Int-1**, followed by the conversion to their keto form **2** to complete this interesting sequence. This reaction mechanism was proved by our own computation for the transition state [7] as well as by experimental employment of the deuterated substrate by the other group [8]. On the basis of this successful as well as convenient proton shift process starting from the CF_3_-containing allylic alcohols **1**, we envisaged its intriguing extension from the synthetic point of view: thus, our idea was that the bisallylic ethers **3** possibly synthesized in a facile manner by way of the *O*-allylation of **1** were recognized as the potential substrates for the Claisen rearrangement as long as the proton shift of **3** to **4** was possible. It is worthwhile to note that, except for our previous report [9], there are no such examples to prepare the compounds **5** by way of [3,3]-sigmatropic rearrangement irrespective of the substituents R^2^, while there are some precedented work on the alkylation route to alternatively get access to compounds like **5** [10,11,12,13]. We thus started our research for the novel utilization of the CF_3_-containing bisallylic ethers **3** as the potent substrates for the Claisen rearrangement via the facile isomerization to the corresponding allyl vinyl ethers **4**.

## 2. Results and Discussion

### 2.1. Preparation of Bisallyl Ethers ***3***

Preparation of the CF_3_-containing allylic alcohols **1** was carried out in a stereoselective fashion following our own developed method like (1) reactions of adequate Grignard reagents with CF_3_CO_2_Et to construct the ketones CF_3_-C(O)-R^1^, (2) their Horner-Wadsworth-Emmons reactions with (EtO)_2_P(O)CH_2_C(O)R^2^ [14], and (3) NaBH_4_ reduction of the resultant α,β-unsaturated ketones. Important to note is the fact that the sequences (1) and (2) could be performed without isolation of the intermediary trifluorinated ketones which allowed the possible formation of **1** even with a “small” R^1^ (for example, **1d** with a Et group as R^1^) whose isolation is usually difficult due to their low boiling points and high volatility [7,15].

Optimization of the reaction conditions for the *O*-allylation was initially performed using the allylic alcohol **1a** as the representative model whose results are summarized in Table 1. First of all, investigation of a base clarified that an excess amount of a 6 M NaOH aqueous solution was the best among tested for the construction of the desired **3a** (Entries 1 to 5). Because BuLi unexpectedly recorded complete recovery (Entry 4), we further changed the reaction temperature or modified the reactivity by the addition of hexamethylphosphoric triamide (HMPA), but both did not give any fruitful results (Entries 15 or 16, respectively). Formation of the saturated ketone **2a** as the byproduct was interpreted as a result of the abstraction of a proton at C^1^ in **1a**, followed by re-protonation at C^3^, and the stronger bases showed a clear tendency to prefer this isomerization except for BuLi. After fixing the base as 6 M NaOH aq., further brief check of a phase transfer catalyst (PTC) pointed out that Bu_4_NI was the reagent of choice (Entries 5 to 7) combined with dichloromethane (DCM) as a solvent (Entries 5, 8, and 9). Entries 10 to 14 were carried out for determination of the best amount of allyl bromide and it was concluded that 2.0 equiv would suffice for our purpose. Final examination on the concentration (Entries 12, 17, and 18), the equiv (Entries 12, 19, and 20) of NaOH, and the reaction period (Entries 12, 22, and 23) led to the final conclusion that the conditions shown in Entry 23 was the best of all for the *O*-allylation of **1a**.

The optimized reaction conditions determined as above were employed for the synthesis of a variety of the CF_3_-containing bisallylic ethers **3** whose results are collected in Table 2. In spite of formation of the side product, ketones **2**, in small amounts, good to excellent isolated yields were attained for the construction of the desired compounds **3** in many instances. However, this was not the case for the substrates **1** possessing alkylsubstituents as R^2^ whose electron-donating effect seemed to lower the acidity of an OH group to slow down the reaction rate and consequently led to recovery of the substrates to some extent (Entries 8 and 9). In our previous report on the 1,3-proton shift of the compounds **1** to the ketones **2** [7], the clear substituent effect for R^2^ was experimentally as well as computationally manifested, and aromatic groups were required for the smooth promotion of this isomerization by effective increase of the acidity of the proton at C^1^. It is important to mention that the clear contrast was pointed out for the substituents R^1^ whose effect was, different from the instance of R^2^, only limited.

### 2.2. Preparation of Allyl Vinyl Ethers ***4*** and One-Pot Isomerization-Claisen Rearrangement from ***3***

The requisite bisallylic ethers **3** in hand, we have at first undertaken the base-promoted isomerization of **3** to **4** (Table 3). A 0.5 equiv of DBU was employed as a base because of its high potency for our previous system to achieve the proton migration of **1** to **2** [7]. As a result, it was observed that the desired isomerization of the bisallyl ether **3a** proceeded in a very smooth fashion at room temperature to furnish the corresponding allyl vinyl ether **4a** as a single stereoisomer (Entry 1). Stereochemistry of **4a** was presumed to be Z on the basis of the fact that the exclusive formation of the (*Z*)-enol silyl ether by the action of lithium diisopropylamide (LDA) to the structurally similar propiophenone was due to the unfavorable sterically repulsive interaction between the methyl and phenyl groups in the corresponding (*E*)-isomer [16]. Because of the attachment of an aromatic group as the R^2^ was essential for the realization of the ready isomerization of **1** to **2**, all compounds **3** employed here possessed benzene-based R^2^ and thus, all of the products **4** were considered to have the same (*Z*)-stereochemistry. Under the room temperature stirring for 4 days, [3,3]-sigmatropic rearrangement of (*Z*)-**4a** simultaneously proceeded and the desired product **5a** was constructed in 16% yield with the 88% *syn* preference (please refer to the Section 2.4. for the explanation of the *syn* selectivity).

Because of the relatively slow reaction rate as described in Entry 1, attempt to raise the reaction temperature was conducted to find out that reflux in THF gave strong influence for the conversion of **3a** which was significantly accelerated with recording almost quantitative combined yields of **4a** and **5a** (Entry 2). Solvent change to toluene provided 45 °C difference for the reflux temperature which seemed to suffice for the sequential isomerization-Claisen rearrangement to produce 91% of the desired product **5a** as a 68:32 diastereomer mixture only in 3 h (Entries 3–4). Necessity of 15 h for **3b** to complete the reaction was interpreted as the destabilization of the transition state by the electron-donating MeO group at the *p*-position of a phenyl moiety in R^1^ (Entry 8). This is in sharp contrast to the case of the substrate **3c** with a fluorine atom at the same position and 3 h reflux in toluene was enough to afford 91% yield of the product **5c** (Entry 11). In both instances of **3b** and **3c**, 40 °C and room temperature reactions alternatively led to the formation of the allyl vinyl ethers **4b** and **4c** in moderate to good yields, respectively (Entries 6 and 10).

The similar substituent effect of R^1^ was clearly understood by comparison of the results in Entries 12–23 with the others, and retardation of the present process was observed by incorporation of alkyl groups for R^1^ but in a less effective manner than the case of R^2^ which completely inhibited the elimination of a proton from C^1^ and thus, no transformation of **1** to **2** was observed [7]. In the case of **3d** with R^1^ = Et, 48 h reflux was necessary for the direct transformation to **5d** which was attained in 87% isolated yield but with the decreased diastereoselectivity to 55:45. Almost similar outcomes were recorded for the substrates **3e** to **3g** with a Ph(CH_2_)_2_ moiety as R^1^ in terms of their chemical yields as well as the stereoselectivities irrespective of variation of the reaction periods.

For acquiring the relationship between the effect of the reaction temperature and time towards the diastereoselectivity, the isolated product **5a** (*syn*:*anti* = 74:26) was submitted to the standard rearrangement conditions using toluene as a solvent (Scheme 2). Thus, reflux for 3 h demonstrated a slight decrease of the isomeric ratio to 68:32 which was further lowered to 66:34 after the prolonged reaction time to 24 h. On the other hand, the initial proportion was completely retained when **5a** was stirred at room temperature, or under reflux without the base, DBU. This brief study proved that a weak base DBU was found to have an ability for epimerization of **5a** at least in part, which was nicely explained by the experimental facts shown in Entries 1, 2, and 4 as well as 7 and 8 in Table 3: the lower the reaction temperature became, the better the diastereomer ratios were obtained. For the instances of the substrates **3** with an alkyl substituent as R^1^, the lower selectivity obtained was similarly understood as a consequence of the requirement of longer reaction times which would offer the higher chance of epimerization at the carbonyl α position in **5**.

### 2.3. Improvement of the Diastereoselectivity of the Claisen Rearrangement Products ***5***

Because it was our interpretation that the low diastereoselectivity of the rearranged products **5** were attributed to the requirement of the harsh conditions like refluxing in toluene at 111 °C, modification of the present system was planned by the addition of appropriate activators for the purpose of lowering the reaction temperature. As described in Table 4, 20 mol% of typical Lewis acids like TiCl_4_, BF_3_·OEt_2_, and AlCl_3_ were independently added to a DCM solution containing **4a** at 0 °C to prove that cleavage of the allyl ether part attached to **1** as shown in Table 2 occurred as the main pathway to yield the ketone **2a** along with a minor quantity of the requisite rearranged product **5a** (Entries 1 to 3, Table 4). Further decrease of the reaction temperature in the case of TiCl_4_ was carried out with the expectation to inhibit this unfavorable route, but **2a** was the sole product obtained even at −80 °C with complete suppression of the construction of **5a** (Entry 4).

The similar consequence was noticed for Sc(OTf)_3_ (Entry 5), and like the cases of other triflates like Mg(OTf)_2_, Yb(OTf)_3_, Gd(OTf)_3_ (not shown in Table 4) [17], the total amounts of fluorinated compounds obtained after the reaction were only in a range of 40% to 60% including **5a** in less than 5% yield. However, interesting to note is the fact that weakly acidic but non-nucleophilic 1,1,1,3,3,3-hexafluoropropan-2-ol (HFIP) [18] promoted the desired transformation nicely to afford **5a** in 66% yield after stirring for 3 days at room temperature (Entry 6).

At the next stage, application of [PdCl_2_·(PhCN)_2_] was implemented on the basis of our previously successful experience on this catalyst for the Ireland-Claisen rearrangement [9,19,20]: as a result, 10 mol% of this catalyst was found to be quite effective for excellent conversion of **4a** to **5a** even at room temperature. After brief examination of the solvent, toluene recorded smooth transformation without forming **2a** and the desired product **5a** was obtained in 70% isolated yield (Entries 7 to 9). The additional benefit was the increase of the diastereoselectivity from 68:32 (Entry 4, Table 3) to 95:5 (Entry 9) where, in line with our expectation, the lowering of the reaction temperature from 111 °C (toluene reflux) to 25 °C would play a significant role at least in part. This pertinent condition was also able to produce the better consequence for both substrates **4b** (R^1^: *p*-MeOC_6_H_4_) and **4c** (R^1^: *p*-FC_6_H_4_), attaining the same level of excellent selectivity in 81% and 65% isolated yields, respectively (Entries 10 and 11). One drawback of this Pd-catalyzed process is the formation of the byproducts **6** in a range of 4 to 8% [21] (Entries 9 to 11) which was not possible to be separated completely by silica gel column chromatography. As described in Scheme 3, subjection of the thermally rearranged product **5a** as a 70:30 diastereomer mixture to this Pd-catalyzed conditions led to the formation of 10% of **6a** (determined by ^19^F NMR), which unambiguously proved that, at least in part, **6a** was obtained as the result of isomerization of **5a**.

### 2.4. Discussion on the Reaction Mechanism

First of all, on the diastereoselectivity of the present Claisen rearrangement, production of the *syn*-isomer was anticipated on the basis of our previous report as shown in Scheme 4 [9]. Thus, Michael addition of butyroyloxazolidinone-based enolate to allyl 4,4,4-trifluorobut-2-enoate, followed by the capture of the resultant enolate by TMSCl furnished the intermediary ketene silyl acetal **Int-2** [22,23]. This intermediate **Int-2** then experienced the Ireland-Claisen rearrangement with the aid of a catalytic amount of [PdCl_2_·(PhCN)_2_] to afford the rearranged product **7** in a highly stereoselective manner along with the unrearranged Michael adduct **8** in 63% and 32% yields, respectively. Stereochemistry of **7** was crystallographically confirmed as 2,3-*anti*,3,4-*syn* [24], the latter of which was conveniently explained by the application of the Cieplak rule [25]. Because incipient transition states (TS) in general are electron-deficient, reactions favorably occur from the face where the better stabilization is accomplished by the electron donation from the adjacent C-C bond (Figure 1). Suppose that the rearrangement occurred from the *si*-face (by way of TS-*si*) in our previous instance, TS σ*^≠^ should be better stabilized by electron-donation from the electronically richer σ_C-R1_ orbital rather than the participation of the electron-deficient σ_C-CF3_. This rule consistently elucidated the preferential formation of the *syn*-isomer as determined by X-ray crystallographic analysis. In the present case, because of the similar [3,3]-sigmatropic rearrangement of the structurally similar substrates with the R-C(CF_3_)H-branched structure at the same position as well as activation by the same catalyst, we believe that the major isomers anticipated for the present case was the *syn* isomers.

In the case of the thermal rearrangement shown in Table 3, there are two distinct groups of **5a**–**c** and **5d**–**g** in terms of the diastereoselectivity obtained which was interpreted in the Section 2.2. as a result of the lower activating character by the electron-donating R^1^ moieties in the latter group, thereby requiring a longer reaction time with possible higher chance of epimerization. Another explanation would be made from the standpoint of steric requirement: the smaller substituents for the latter group as R^1^ (Et and PhCH_2_CH_2_ with the revised Taft Es values (Es’) [26] of 0.08 and 0.35, respectively) than the case of the former (for example, 2.31 and 0.78 Es’ values for Ph and CF_3_, respectively: the bigger numbers indicate the more steric bulkiness) would alleviate the unfavorable steric factors for the approach from the electronically less favorable *re*-face, thereby the diastereoselectivity was more or less lowered.

## 3. Conclusions

As depicted above, we have succeeded in the development of a new route of the Claisen rearrangement starting from the base-mediated isomerization of the bisallylic alcohols **3** to the allyl vinyl ethers **4**, and the following formation of the rearranged products **5** were found to be realized in a one-pot manner when the reactions were conducted under toluene reflux conditions. An excellent alternative method was to treat **4** with such a catalyst as [PdCl_2_·(PhCN)_2_] in toluene, and in spite of contamination by a small amount of the isomerized products **6**, the highest diastereomeric ratio of 95:5 was recorded which are well compared with the ones of 68:32 obtained by the direct thermal rearrangement from **3a**. The weakly acidic HFIP was another choice for the present process which, in spite of requirement of 3 days for completion, the reaction nicely proceeded with recording good yield as well as stereoselectivity.

## 4. Materials and Methods

### 4.1. General Information

Unless otherwise noted, materials were obtained from commercial suppliers and were used without further purification. All manipulations involving air-sensitive materials were performed under argon. Anhydrous Et_2_O, THF and DCM were purchased and were used without further purification.

^1^H, ^13^C, and ^19^F NMR spectra were recorded with a JEOL JNM-LA300 (^1^H: 300 MHz, ^13^C: 75 MHz, and ^19^F: 283 MHz) in CDCl_3_. Chemical shifts were recorded in parts per million (ppm), downfield from internal tetramethylsilane (for ^1^H NMR, Me_4_Si: δ 0.00 ppm, ^13^C NMR, CDCl_3_: 77.0 ppm for the center peak, and for ^19^F NMR, C_6_F_6_: δ −163.0 ppm). ^13^C NMR spectra of minor isomers may not be fully reported due to difficult visualization of peaks with small intensities even after a long data acquisition time. Please refer to the Appendix A for the copies of ^1^H and ^13^C NMR charts for new compounds. Infrared (IR) spectra were obtained on a JASCO FT/IR-4100 spectrometer, and all spectra were reported in wave numbers (cm^−1^). High resolution mass spectra in a FAB mode were acquired on a JEOL JMS-700. Analytical thin-layer chromatography (TLC) on silica gel 60 F_254_ (Merck, Kenilworth, NJ, USA) was routinely used for monitoring reactions usually using a mixture of hexane (Hex, Cape Town, South Africa) and ethyl acetate (AcOEt) or DCM. Column chromatography was conducted with silica gel 60 N (spherical, neutral, 63–210 nm, Kanto, Tokyo, Japan).

### 4.2. General Procedure for the Preparation of Bisallyl Ethers

#### 4.2.1. (*E*)-1,1,1-Trifluoro-2,4-diphenyl-4-{(prop-2-en-1-yl)oxy}but-2-ene (**3a**)

2 mL of 6 *M* NaOH aq. (12.0 mmol) was added to a 30 mL round-bottomed flask containing 0.14 g of (*E*)-4,4,4-trifluoro-1,3-diphenylbut-2-en-1-ol **1a** (0.50 mmol), 0.12 g of allyl bromide (1.0 mmol), 0.018 g of tetrabutyl-ammonium iodide (0.050 mmol), and 5 mL of DCM, and the whole solution was stirred for 48 h at room temperature. After quenching by 1.5 mL of 6 *M* HCl aq., the reaction mixture was extracted by DCM three times which was dried over anhydrous Na_2_SO_4_. Filtration of the desiccant and evaporation of the volatiles furnished a crude mixture which was chromatographed with silica-gel using Hex:AcOEt = 6:1 as an eluent to afford 0.16 g of the title compound (0.49 mmol) in 98% yield as a colorless oil.

Rf = 0.71 (Hex:AcOEt = 6:1). ^1^H NMR: δ 3.79 (1H, ddt, *J* = 12.6, 6.0, 1.5 Hz), 3.86 (1H, ddt, *J* = 12.6, 5.7, 1.5 Hz), 4.75 (1H, d, *J* = 9.3 Hz), 5.11 (1H, dq, *J* = 10.2, 1.5 Hz), 5.17 (1H, dq, *J* = 17.1, 1.8 Hz), 5.82 (1H, ddt, *J* = 17.4, 10.3, 5.4 Hz), 6.57 (1H, dq, *J* = 9.3, 1.5 Hz), 7.20–7.44 (10H, m). ^13^C NMR: δ 69.1, 76.7, 117.4, 123.1 (q, *J* = 272.9 Hz), 126.9, 128.2, 128.5, 128.7, 128.9, 129.7, 131.5, 132.5 (q, *J* = 30.4 Hz), 134.1, 135.9 (q, *J* = 5.6 Hz), 139.6. ^19^F NMR: δ −67.77 (s). IR (neat): ν 3064, 3031, 2861, 1494, 1454, 1317, 1254, 1174, 1124, 1064, 702. HRMS (FAB+, *m*/*z*): [M + H]^+^ calcd. for C_19_H_18_F_3_O, 319.1304, found 319.1320.

#### 4.2.2. (*E*)-1,1,1-Trifluoro-2-(4-methoxyphenyl)-4-phenyl-4-{(prop-2-en-1-yl)oxy}but-2-ene (**3b**)

Instead of (*E*)-4,4,4-trifluoro-1,3-diphenylbut-2-en-1-ol **1a**, 0.16 g of (*E*)-4,4,4-trifluoro-3-(4-methoxyphenyl)-1-phenylbut-2-en-1-ol **1b** (0.51 mmol) was employed and 0.15 g of the title compound (0.43 mmol) was isolated in 83% yield as a colorless oil.

Rf = 0.37 (Hex:AcOEt = 6:1). ^1^H NMR: δ 3.76–3.90 (2H, m), 3.86 (3H, s), 4.78 (1H, d, *J* = 9.0 Hz), 5.10–5.21 (2H, m), 5.84 (1H, ddt, *J* = 17.1, 10.2, 5.4 Hz), 6.55 (1H, dq, *J* = 9.6, 1.5 Hz), 6.95 (2H, dt, *J* = 8.7, 1.8 Hz), 7.16–7.26 (4H, m), 7.30–δ 7.38 (3H, m). ^13^C NMR: δ 55.2, 69.1, 76.8, 113.9, 117.3, 123.2 (q, *J* = 272.9 Hz), 123.5, 126.9, 128.2, 128.7, 130.9, 132.2 (q, *J* = 30.4 Hz), 134.2, 135.7 (q, *J* = 5.0 Hz), 139.7, 160.0. ^19^F NMR: δ −67.93 (s). IR (neat): ν 3008, 2936, 2840, 1609, 1515, 1455, 1251, 1124, 927, 835, 700. HRMS (FAB+, *m*/*z*): [M + H]^+^ calcd. for C_20_H_20_F_3_O_2_, 349.1410, found 349.1381.

#### 4.2.3. (*E*)-1,1,1-Trifluoro-2-(4-fluorophenyl)-4-phenyl-4-{(prop-2-en-1-yl)oxy}but-2-ene (**3c**)

Instead of (*E*)-4,4,4-trifluoro-1,3-diphenylbut-2-en-1-ol **1a**, 0.15 g of (*E*)-4,4,4-trifluoro-3-(4-fluorophenyl)-1-phenylbut-2-en-1-ol **1c** (0.51 mmol) was employed and 0.14 g of the title compound (0.42 mmol) was isolated in 84% yield as a colorless oil.

Rf = 0.57 (Hex:AcOEt = 6:1). ^1^H NMR: δ 3.76–3.87 (2H, m), 4.71 (1H, d, *J* = 9.3 Hz), 5.10–5.20 (2H, m), 5.82 (1H, ddt, *J* = 17.1, 10.5, 5.7 Hz), 6.60 (1H, dq, *J* = 9.3, 1.8 Hz), 7.12 (2H, tt, *J* = 8.7, 2.1 Hz), 7.18–7.25 (4H, m), 7.29–7.39 (3H, m). ^13^C NMR: δ 69.1, 76.8, 115.7 (q, *J* = 21.1 Hz), 117.4, 122.9 (q, *J* = 272.9 Hz), 126.9, 127.4 (d, *J* = 3.1 Hz), 128.4, 128.8, 131.4 (q, *J* = 30.4 Hz), 131.6 (d, *J* = 8.7 Hz), 134.1, 136.5 (q, *J* = 5.0 Hz), 139.4, 163.1 (d, *J* = 248.8 Hz). ^19^F NMR: δ −67.96 (3F, s), −113.33~113.43 (1F, m). IR (neat): ν 3031, 2862, 1605, 1513, 1455, 1317, 1236, 1175, 1124, 842, 700. HRMS (FAB+, *m*/*z*): [M + H]^+^ calcd. for C_19_H_17_F_4_O, 337.1216, found 337.1228.

#### 4.2.4. (*E*)-1-Phenyl-1-{(prop-2-en-1-yl)oxy}-3-(trifluoromethyl)pent-2-ene (**3d**)

Instead of (*E*)-4,4,4-trifluoro-1,3-diphenylbut-2-en-1-ol **1a**, 0.12 g of (*E*)-1-phenyl-3-(trifluoromethyl)pent-2-en-1-ol **1d** (0.50 mmol) was employed and 0.12 g of the title compound (0.45 mmol) was isolated in 90% yield as a colorless oil.

Rf = 0.34 (Hex:AcOEt = 10:1). ^1^H NMR: δ 1.09 (3H, t, *J* = 7.8 Hz), 2.27–2.40 (2H, m), 3.89–4.01 (2H, m), 5.11 (1H, d, *J* = 8.7 Hz), 5.22 (1H, dq, *J* = 10.2, 1.8 Hz), 5.27 (1H, dq, *J* = 17.1, 1.5 Hz), 5.92 (1H, ddt, *J* = 17.4, 10.5, 5.7 Hz), 6.26 (1H, dq, *J* = 8.7, 1.5 Hz), 7.26–7.40 (5H, m). ^13^C NMR: δ 13.5, 19.5, 69.2, 76.0, 117.5, 124.3 (q, *J* = 273.5 Hz), 126.9, 128.2, 128.8, 132.6 (q, *J* = 27.9 Hz), 133.7 (q, *J* = 5.6 Hz), 134.3, 139.9. ^19^F NMR: δ −68.54 (s). IR (neat): ν 2982, 2944, 2884, 1731, 1454, 1322, 1252, 1177, 1063, 926, 700. HRMS (FAB+, *m*/*z)*: [M]^+^ calcd. for C_15_H_17_F_3_O, 270.1226, found 270.1206.

#### 4.2.5. (*E*)-1,5-Diphenyl-1-{(prop-2-en-1-yl)oxy}-3-(trifluoromethyl)pent-2-ene (**3e**)

Instead of (*E*)-4,4,4-trifluoro-1,3-diphenylbut-2-en-1-ol **1a**, 0.16 g of (*E*)-1,5-diphenyl-3-(trifluoromethyl)pent-2-en-1-ol **1e** (0.52 mmol) was employed and 0.15 g of the title compound (0.42 mmol) was isolated in 81% yield as a colorless oil.

Rf = 0.63 (Hex:AcOEt = 6:1). ^1^H NMR: δ 2.55–2.82 (4H, m), 3.82–3.83 (1H, m), 3.835–3.844 (1H, m), 4.97 (1H, d, *J* = 9.0 Hz), 5.18–5.29 (2H, m), 5.89 (1H, ddt, *J* = 17.1, 10.2, 5.7 Hz), 6.35 (1H, dq, *J* = 8.7, 1.2 Hz), 7.18–7.40 (10H, m). ^13^C NMR: δ 28.6, 34.8, 69.1, 76.2, 117.5, 124.2 (q, *J* = 273.6 Hz), 126.4, 127.0, 128.3, 128.4, 128.6, 128.8, 130.3 (q, *J* = 27.9 Hz), 134.3, 135.3 (q, *J* = 6.3 Hz), 139.7, 140.8. ^19^F NMR: δ −67.96 (s). IR (neat): ν 3029, 3012, 2871, 1495, 1455, 1326, 1218, 1165, 1120, 765, 700. HRMS (FAB+, *m*/*z*): [M + H]^+^ calcd. for C_21_H_22_F_3_O, 347.1617, found 347.1631.

#### 4.2.6. (*E*)-1-(4-Methoxyphenyl)-5-phenyl-1-{(prop-2-en-1-yl)oxy}-3-(trifluoromethyl)-pent-2-ene (**3f**)

Instead of (*E*)-4,4,4-trifluoro-1,3-diphenylbut-2-en-1-ol **1a**, 0.17 g of (*E*)-1-(4-methoxyphenyl)-5-phenyl-3-(trifluoromethyl)pent-2-en-1-ol **1f** (0.50 mmol) was employed and 0.14 g of the title compound (0.36 mmol) was isolated in 71% yield as a colorless oil.

Rf = 0.57 (Hex:AcOEt = 6:1). ^1^H NMR: δ 2.47–2.84 (4H, m), 3.79–3.82 (2H, m), 3.81 (3H, s), 4.91 (1H, d, *J* = 8.7 Hz), 5.17–5.28 (2H, m), 5.88 (1H, ddt, *J* = 17.1, 10.2, 5.7 Hz), 6.36 (1H, dq, *J* = 8.4, 1.5 Hz), 6.86–6.91 (2H, m), 7.18–7.34 (7H, m). ^13^C NMR: δ 28.5, 34.7, 55.2, 68.9, 75.6, 114.1, 117.5, 124.2 (q, *J* = 273.5 Hz), 126.3, 128.32, 128.34, 128.6, 129.8 (q, *J* = 27.9 Hz), 131.7, 134.3, 135.0 (q, *J* = 5.6 Hz), 140.8, 159.6. ^19^F NMR: δ −67.99 (s). IR (neat): ν 3009, 2936, 2839, 1611, 1512, 1326, 1253, 1119, 833, 760, 700. HRMS (FAB+, *m*/*z*): [M + H]^+^ calcd. for C_22_H_24_F_3_O_2_, 377.1723, found 377.1745.

#### 4.2.7. (*E*)-1-(4-Bromophenyl)-5-phenyl-1-{(prop-2-en-1-yl)oxy}-3-(trifluoromethyl)pent-2-ene (**3g**)

Instead of (*E*)-4,4,4-trifluoro-1,3-diphenylbut-2-en-1-ol **1a**, 0.19 g of (*E*)-1-(4-bromophenyl)-5-phenyl-3-(trifluoromethyl)pent-2-en-1-ol **1g** (0.51 mmol) was employed and 0.19 g of the title compound (0.44 mmol) was isolated in 87% yield as a colorless oil.

Rf = 0.49 (Hex:AcOEt = 10:1). ^1^H NMR: δδ 2.53–2.87 (4H, m), 3.79–3.82 (2H, m), 4.88 (1H, d, *J* = 8.7 Hz), 5.21 (1H, dq, *J* = 10.2, 1.5 Hz), 5.24 (1H, dq, *J* = 17.4, 1.5 Hz), 5.87 (1H, ddt, *J* = 17.1, 10.2, 5.7 Hz), 6.26 (1H, dq, *J* = 8.4, 1.2 Hz), 7.08–7.12 (2H, m), 7.18–7.35 (5H, m), 7.45–7.50 (2H, m). ^13^C NMR: δ28.5, 34.7, 69.2, 75.4, 117.7, 122.2, 124.1 (q, *J* = 274.2 Hz), 126.4, 128.4, 128.59, 128.63, 130.8 (q, *J* = 28.5 Hz), 131.9, 134.1, 134.7 (q, *J* = 6.9 Hz), 138.7, 140.6. ^19^F NMR: δ −67.90 (s). IR (neat): ν 3011, 2866, 1590, 1487, 1325, 1164, 1119, 1071, 1011, 762, 700. HRMS (FAB+, *m*/*z*): [M + H]^+^ calcd. for C_21_H_21_BrF_3_O, 425.0722, found 425.0757.

### 4.3. General Procedure for the Preparation of Ally Vinyl Ethers

#### 4.3.1. (*E*)-4,4,4-Trifluoro-1,3-diphenyl-1-{(prop-2-en-1-yl)oxy}but-1-ene (**4a**)

In a two-necked 30 mL round-bottomed flask were added under an argon atmosphere 0.16 g of (*E*)-1,1,1-trifluoro-2,4-diphenyl-4-(prop-2-en-1-yloxy)but-2-ene **3a** (0.50 mmol), 0.039 g of DBU (0.25 mmol), and THF (5.0 mL), and the whole mixture was stirred for 96 h at room temperature. After quenching the reaction by the addition of H_2_O and usual workup, the crude material was purified by silica-gel chromatography using Hex:DCM = 10:1 as an eluent to furnish 0.12 g (0.36 mmol) of the title compound as a colorless oil in 73% yield.

Rf = 0.43 (Hex:DCM = 6:1). ^1^H NMR: δ4.02 (1H, ddt, *J* = 12.8, 5.9, 1.2 Hz), 4.14 (1H, ddt, *J* = 12.8, 5.9, 1.2 Hz), 4.74 (1H, quint, *J* = 9.8 Hz), 5.15 (1H, dq, *J* = 10.2, 1.2 Hz), 5.21 (1H, dq, *J* = 17.1, 1.5 Hz), 5.57 (1H, d, *J* = 9.9 Hz), 5.86 (1H, ddt, *J* = 17.1, 10.5, 5.7 Hz), 7.25–7.47 (10H, m). ^13^C NMR: δ 46.5 (q, *J* = 28.5 Hz), 71.1, 107.2 (q, *J* = 2.5 Hz), 117.7, 126.3 (q, *J* = 279.1 Hz), 126.8, 127.9, 128.5, 128.6, 128.91, 128.93, 133.4, 134.9, 135.9 (q, *J* = 1.3 Hz), 157.2. ^19^F NMR: δ −70.58 (d, *J* = 9.3 Hz). IR (neat): ν 3033, 2931, 1654, 1495, 1251, 1165, 1111, 1052, 930, 771, 700. HRMS (FAB+, *m*/*z*): [M + H]^+^ calcd. for C_19_H_18_F_3_O, 319.1304, found 319.1313.

#### 4.3.2. (*E*)-4,4,4-Trifluoro-3-(4-methoxyphenyl)-1-phenyl-1-{(prop-2-en-1-yl)oxy}but-1-ene (**4b**)

Instead of (*E*)-1,1,1-trifluoro-2,4-diphenyl-4-{(prop-2-en-1-yl)oxy}but-2-ene **3a**, 0.18 g of (*E*)-1,1,1-trifluoro-2-(4-methoxyphenyl)-4-phenyl-4-{(prop-2-en-1-yl)oxy}but-2-ene **3b** (0.50 mmol) was employed and stirring was continued for 48 h at 40 °C to furnish 0.091 g of the title compound (0.26 mmol) was isolated in 52% yield as a colorless oil.

Rf = 0.37 (Hex:DCM = 3:1). ^1^H NMR: δ 3.80 (3H, s), 4.03 (1H, ddt, *J* = 12.9, 5.7, 1.2 Hz), 4.14 (1H, ddt, *J* = 12.6, 5.7, 1.2 Hz), 4.69 (1H, quint, *J* = 9.6 Hz), 5.17 (1H, dq, *J* = 10.2, 1.2 Hz), 5.23 (1H, dq, *J* = 17.1, 1.5 Hz), 5.56 (1H, d, *J* = 9.6 Hz), 5.87 (1H, ddt, *J* = 17.3, 10.5, 5.7 Hz), 6.86–6.96 (2H, m), 7.30–7.40 (5H, m), 7.42–7.47 (2H, m). ^13^C NMR: δ 45.6 (q, *J* = 29.2 Hz), 55.1, 71.1, 107.4 (q, *J* = 2.5 Hz), 114.0, 117.7, 126.4 (q, *J* = 279.8 Hz), 126.7, 127.8 (q, *J* = 1.9 Hz), 128.5, 128.8, 129.9, 133.4, 134.8, 156.9, 159.2. ^19^F NMR: δ −71.07 (d, *J* = 9.0 Hz). IR (neat): ν 3061, 2934, 1613, 1514, 1251, 1163, 1110, 1036, 992, 828, 700. HRMS (FAB+, *m*/*z*): [M + H]^+^ calcd. for C_20_H_20_F_3_O_2_, 349.1410, found 349.1451.

#### 4.3.3. (*E*)-4,4,4-Trifluoro-3-(4-fluorophenyl)-1-phenyl-1-{(prop-2-en-1-yl)oxy}but-1-ene (**4c**)

Instead of (*E*)-1,1,1-trifluoro-2,4-diphenyl-4-{(prop-2-en-1-yl)oxy}but-2-ene **3a**, 0.17 g of (*E*)-1,1,1-trifluoro-2-(4-fluorophenyl)-4-phenyl-4-{(prop-2-en-1-yl)oxy}but-2-ene **3c** (0.50 mmol) was employed and stirring was continued for 96 h at room temperature to furnish 0.14 g of the title compound (0.42 mmol) was isolated in 84% yield as a colorless oil.

Rf = 0.37 (Hex:DCM = 6:1). ^1^H NMR: δ 4.03 (1H, ddt, *J* = 12.6, 5.7, 1.5 Hz), 4.14 (1H, ddt, *J* = 12.6, 5.7, 1.5 Hz), 4.72 (1H, quint, *J* = 9.5 Hz), 5.17 (1H, dq, *J* = 10.8, 1.5 Hz), 5.21 (1H, dq, *J* = 17.1, 1.5 Hz), 5.52 (1H, d, *J* = 9.6 Hz), 5.85 (1H, ddt, *J* = 17.1, 10.2, 5.7 Hz), 7.01–7.08 (2H, m), 7.34–7.38 (5H, m), 7.39–7.46 (2H, m).^13^C NMR: δ 45.7 (q, *J* = 27.9 Hz), 71.0, 106.8 (q, *J* = 1.9 Hz), 115.5 (d, *J* = 21.7 Hz), 117.9, 126.2 (q, *J* = 279.1 Hz), 126.8, 128.5, 129.0, 130.5 (d, *J* = 8.1 Hz), 131.7 (q, *J* = 1.8 Hz), 133.2, 134.7, 157.4, 162.4 (d, *J* = 246.2 Hz). ^19^F NMR: δ −71.01 (3F, d, *J* = 9.0 Hz), −115.74~−115.66 (1F, m). IR (neat): ν 3084, 2935, 1655, 1607, 1512, 1251, 1167, 1112, 1052, 832, 699. HRMS (FAB+, *m*/*z*): [M]^+^ calcd. for C_19_H_16_F_4_O, 336.1132, found 336.1164.

#### 4.3.4. (*E*)-1-Phenyl-1-{(prop-2-en-1-yl)oxy}-3-(trifluoromethyl)pent-1-ene (**4d**)

Instead of (*E*)-1,1,1-trifluoro-2,4-diphenyl-4-{(prop-2-en-1-yl)oxy}but-2-ene **3a**, 0.14 g of (*E*)-3-(trifluoromethyl)-1-phenyl-1-{(prop-2-en-1-yl)oxy}pent-2-ene **3d** (0.50 mmol) was employed and stirring was continued for 48 h with refluxing to furnish 0.08 g of the title compound (0.30 mmol) was isolated in 59% yield as a colorless oil.

Rf = 0.49 (Hex:DCM = 6:1). ^1^H NMR: δ 0.98 (3H, t, *J* = 7.5 Hz), 1.40-1.53 (1H, m), 1.79–1.93 (1H, m), 3.34–3.51 (1H, m), 4.09–4.20 (2H, m), 4.99 (1H, d, *J* = 9.9 Hz), 5.21 (1H, dq, *J* = 10.5, 1.2 Hz), 5.27 (1H, dq, *J* = 17.1, 1.5 Hz), 5.95 (1H, ddt, *J* = 17.1, 10.5, 5.7 Hz), 7.34–7.39 (3H, m), 7.45–7.48 (2H, m). ^13^C NMR: δ11.3, 21.7, 42.0 (q, *J* = 26.6 Hz), 71.2, 107.7 (q, *J* = 2.5 Hz), 117.6, 126.7, 127.2 (q, *J* = 279.1 Hz), 128.5, 128.7, 133.5, 135.1, 158.1. ^19^F NMR: δ −71.98 (d, *J* = 9.3 Hz). IR (neat): ν 2972, 2880, 1659, 1323, 1254, 1173, 1121, 1068, 997, 922, 698. HRMS (FAB+, *m*/*z*): [M]^+^ calcd. for C_15_H_17_F_3_O, 270.1226, found 270.1236.

#### 4.3.5. (*E*)-1,5-Diphenyl-1-{(prop-2-en-1-yl)oxy}-3-(trifluoromethyl)pent-1-ene (**4e**)

Instead of (*E*)-1,1,1-trifluoro-2,4-diphenyl-4-{(prop-2-en-1-yl)oxy}but-2-ene **3a**, 0.17 g of (*E*)-1,5-diphenyl-1-{(prop-2-en-1-yl)oxy}-3-(trifluoro-methyl)pent-2-ene **3e** (0.50 mmol) was employed and stirring was continued for 24 h with refluxing to furnish 0.11 g of the title compound (0.31 mmol) was isolated in 62% yield as a colorless oil.

Rf = 0.43 (Hex:DCM = 6:1). ^1^H NMR: δ1.71–1.84 (1H, m), 2.05–2.18 (1H, m), 2.57–2.80 (2H, m), 3.47–3.64 (1H, m), 4.10 (1H, ddt, *J* = 12.6, 5.7, 1.2 Hz), 4.17 (1H, ddt, *J* = 12.6, 5.4, 1.2 Hz), 5.05 (1H, d, *J* = 10.2 Hz), 5.17–5.29 (2H, m), 5.91 (1H, ddt, *J* = 17.1, 10.5, 5.7Hz), 7.16–7.22 (3H, m), 7.26–7.32 (2H, m), 7.35–7.42 (3H, m), 7.46–7.49 (2H, m). ^13^C NMR: δ 30.4 (q, *J* = 1.9 Hz), 32.8, 40.2 (q, *J* = 26.6 Hz), 71.1, 107.4 (q, *J* = 2.5 Hz), 117.5, 126.0, 126.8, 127.1 (q, *J* = 279.1 Hz), 128.39, 128.42, 128.5, 128.8, 133.5, 135.0, 141.3, 158.3. ^19^F NMR: δ −71.93 (d, *J* = 9.0 Hz). IR (neat): ν 3029, 2931, 1658, 1496, 1455, 1255, 1164, 1114, 932, 772, 698. HRMS (FAB+, *m*/*z*): [M]^+^ calcd. for C_21_H_21_F_3_O, 346.1539, found 346.1546.

#### 4.3.6. (*E*)-1-(4-Methoxyphenyl)-5-phenyl-1-{(prop-2-en-1-yl)oxy}-3-(trifluoromethyl)-pent-1-ene (**4f**)

Instead of (*E*)-1,1,1-trifluoro-2,4-diphenyl-4-{(prop-2-en-1-yl)oxy}but-2-ene **3a**, 0.19 g of (*E*)-1-(4-methoxyphenyl)-5-phenyl-1-{(prop-2-en-1-yl)oxy}-3-(trifluoromethyl)pent-2-ene **3f** (0.50 mmol) was employed and stirring was continued for 48 h with refluxing to furnish 0.079 g of the title compound (0.21 mmol) was isolated in 42% yield as a colorless oil.

Rf = 0.40 (Hex:DCM = 6:1). ^1^H NMR: δ 1.70–1.83 (1H, m), 2.06–2.18 (1H, m), 2.56–2.66 (1H, m), 3.45–3.62 (1H, m), 3.84 (3H, s), 4.09 (1H, ddt, *J* = 12.6, 5.4, 1.5 Hz), 4.16 (1H, ddt, *J* = 12.6, 5.4, 1.5 Hz), 4.95 (1H, d, *J* = 10.2 Hz), 5.18 (1H, dq, *J* = 10.5, 1.8 Hz), 5.26 (1H, dq, *J* = 17.4, 1.8 Hz), 5.91 (1H, ddt, *J* = 17.3, 10.4, 5.7 Hz), 6.89–6.93 (2H, m), 7.16–7.21 (3H, m), 7.26–7.31 (2H, m), 7.38–7.43 (2H, m). ^13^C NMR: δ 30.5 (q, *J* = 1.3 Hz), 32.8, 40.2 (q, *J* = 26.6 Hz), 55.3, 71.1, 105.9 (q, *J* = 2.5 Hz), 113.8, 117.4, 126.0, 127.2 (q, *J* = 279.1 Hz), 127.4, 128.1, 128.36, 128.42, 133.6, 141.4, 158.0, 160.1. ^19^F NMR: δ −72.01 (d, *J* = 9.0 Hz). IR (neat): ν 3030, 2954, 1608, 1511, 1291, 1253, 1111, 1034, 932, 840, 699. HRMS (FAB+, *m*/*z*): [M + H]^+^ calcd. for C_22_H_24_F_3_O_2_, 377.1723, found 377.1728.

#### 4.3.7. (E)-1-(4-Bromophenyl)-5-phenyl-1-{(prop-2-en-1-yl)oxy}-3-(trifluoromethyl)pent-1-ene (**4g**)

Instead of (*E*)-1,1,1-trifluoro-2,4-diphenyl-4-{(prop-2-en-1-yl)oxy}but-2-ene **3a**, 0.21 g of (*E*)-1-(4-bromophenyl)-5-phenyl-1-{(prop-2-en-1-yl)oxy}-3-(trifluoromethyl)pent-2-ene **3g** (0.50 mmol) was employed and stirring was continued for 96 h at room temperature to furnish 0.21 g of the title compound (0.21 mmol) was isolated in 99% yield as a colorless oil.

Rf = 0.46 (Hex:DCM = 6:1). ^1^H NMR: δ 1.71–1.84 (1H, m), 2.07–2.22 (1H, m), 2.56–2.82 (2H, m), 3.48–3.62 (1H, m), 4.07 (1H, ddt, *J* = 12.6, 5.4, 1.2 Hz), 4.14 (1H, ddt, *J* = 12.9, 5.7, 1.2 Hz), 5.05 (1H, d, *J* = 9.9 Hz), 5.20 (1H, dq, *J* = 10.2, 1.5 Hz), 5.24 (1H, dq, *J* = 17.1, 1.5 Hz), 5.89 (1H, ddt, *J* = 17.3,10.4, 5.7 Hz), 7.17–7.22 (3H, m), 7.27–7.36 (4H, m), 7.50–7.54 (2H, m). ^13^C NMR: δ 30.3 (q, *J* = 1.8 Hz), 32.8, 40.3 (q, *J* = 27.3 Hz), 71.3, 108.2 (q, *J* = 2.5 Hz), 117.7, 122.9, 126.1, 127.0 (q, *J* = 279.7 Hz), 128.3, 128.38, 128.41, 131.7, 133.2, 133.9, 141.1, 157.3. ^19^F NMR: δ −71.86 (d, *J* = 9.3 Hz). IR (neat): ν 3029, 2931, 2871, 1658, 1486, 1330, 1255, 1169, 933, 822, 699. HRMS (FAB+, *m*/*z*): [M]^+^ calcd. for C_21_H_20_F_3_O, 424.0644, found 424.0661.

### 4.4. General Procedure for the Claisen Rearrangement of Ally Vinyl Ethers

#### 4.4.1. 4,4,4-Trifluoro-1,3-diphenyl-2-(prop-2-en-1-yl)butan-1-one (**5a**)

##### Method 1. By Heating (Isomerization-Rearrangement)

In a two-necked 30 mL round-bottomed flask were added under an argon atmosphere 0.16 g of (*E*)-1,1,1-trifluoro-2,4-diphenyl-4-{(prop-2-en-1-yl)oxy}but-2-ene **3a** (0.50 mmol), 0.039 g of DBU (0.25 mmol), and toluene (5.0 mL), and the whole mixture was refluxed for 3 h. After quenching the reaction by the addition of H_2_O and usual workup, the crude material was purified by silica-gel chromatography using Hex:AcOEt = 10:1 as an eluent to furnish 0.15 g (0.46 mmol) of an inseparable 68:32 diastereomer mixture of the title compound as a colorless oil in 91% yield.

##### Method 2. Rearrangement of Enol Ethers with the Aid of a Palladium Catalyst

In a two-necked 30 mL round-bottomed flask were added under an argon atmosphere 0.16 g of (*E*)-4,4,4-trifluoro-1,3-diphenyl-1-(prop-2-en-1-yloxy)but-1-ene **4a** (0.50 mmol), 0.019 g of [PdCl_2_(PhCN)_2_] (0.05 mmol), and toluene (5.0 mL), and the whole mixture was stirred for 5 h at room temperature. After passing short-path chromatography, the mixture was purified by silica-gel chromatography using Hex:DCM = 6:1 as an eluent to furnish 0.11 g (0.35 mmol) of an inseparable 95:5 diastereomer mixture of the title compound as a colorless oil in 70% yield.

Rf = 0.40(Hex:AcOEt = 10:1). IR (neat): ν 3066, 2956, 1683, 1596, 1448, 1254, 1165, 1120, 1001, 923, 702. HRMS (FAB+, *m*/*z*): [M]^+^ calcd. for C_19_H_17_F_3_O, 318.1226, found 318.1259.

##### Major Isomer

^1^H NMR: δ 1.98–2.06 (1H, m), 2.15–2.27 (1H, m), 3.99 (1H, dq, *J* = 10.8, 8.7 Hz), 4.24 (1H, td, *J* = 10.4, 3.6 Hz), 4.72–4.83 (2H, m), 5.43 (1H, dddd, *J* = 16.2, 10.2, 7.5, 6.6 Hz), 7.40–8.04 (10H, m). ^13^C NMR: δ 36.1, 44.0 (q, *J* = 1.2 Hz), 51.8 (q, *J* = 25.4 Hz), 118.1, 126.5 (q, *J* = 281.0 Hz), 128.3, 128.5, 128.7, 128.8, 132.7 (q, *J* = 1.8 Hz), 133.1, 133.3, 137.4, 201.4. ^19^F NMR: δ −67.81 (d, *J* = 9.0 Hz).

##### Minor Isomer

^1^H NMR: δ 2.70 (1H, t, *J* = 6.6 Hz), 3.84–3.94 (1H, m), 4.31–4.38 (H, m), 4.96–5.09 (2H, m), 5.58–5.72 (1H, m), 7.15–8.04 (10H, m). ^13^C NMR: δ 35.4 (q, *J* = 1.8 Hz), 47.1, 50.9 (q, *J* = 26.1 Hz), 118.4, 127.0 (q, *J* = 279.8 Hz), 128.0, 128.1, 128.37, 128.43, 129.0, 133.0, 133.3, 134.0 (q, *J* = 2.5 Hz), 137.3, 200.3. ^19^F NMR: δ −64.45 (d, *J* = 9.0 Hz).

The byproduct possibly (*E*)-4,4,4-trifluoro-1,3-diphenyl-2-(prop-1-en-1-yl)butan-1-one (**6a**) as a 73:27 diastereomer mixture was observed as an inseparable mixture with **5a** whose representative NMR data were described below.

^1^H NMR: δ 1.38 (3H, dd, *J* = 6.5, 1.5 Hz), 4.28 (1H, dq, *J* = 10.5, 8.7 Hz), 4.73 (1H, t, *J* = 9.8 Hz), 4.94 (1H, ddq, *J* = 15.7, 9.2, 1.7 Hz), 5.46 (1H, dq, *J* = 15.3, 6.6 Hz), 7.17–8.04 (10H, m). ^13^C NMR: δ 17.8, 49.3 (d, *J* = 1.2 Hz), 50.9 (q, *J* = 25.6 Hz), 126.0, 128.1, 128.37, 128.44, 128.7, 130.3, 131.9, 133.3, 198.3. ^19^F NMR: δ −64.76 (d, *J* = 9.9 Hz; **minor**), −68.59 (d, *J* = 9.0 Hz; **major**).

#### 4.4.2. 4,4,4-Trifluoro-3-(4-methoxyphenyl)-1-phenyl-2-(prop-2-en-1-yl)butan-1-one (**5b**)

##### Method 1

Instead of (*E*)-1,1,1-trifluoro-2,4-diphenyl-4-{(prop-2-en-1-yl)oxy}but-2-ene **3a**, 0.18 g of (*E*)-1,1,1-trifluoro-2-(4-methoxyphenyl)-4-phenyl-4-{(prop-2-en-1-yl)oxy}but-2-ene **3b** (0.50 mmol) was employed and stirring was continued for 15 h under reflux to furnish 0.16 g (0.45 mmol) of an inseparable 66:34 diastereomer mixture of the title compound was isolated in 88% yield as a colorless oil.

##### Method 2

Instead of (*E*)-4,4,4-trifluoro-1,3-diphenyl-1-(prop-2-en-1-yloxy)but-1-ene **4a**, 0.17 g of (*E*)-4,4,4-trifluoro-3-(4-methoxyphenyl)-1-phenyl-1-{(prop-2-en-1-yl)oxy}but-1-ene **4b** (0.50 mmol) was employed and stirring was continued for 5 h to furnish 0.14 g (0.40 mmol) of an inseparable 95:5 diastereomer mixture of the title compound was isolated in 81% yield as a colorless oil.

Rf = 0.31 (Hex:AcOEt = 6:1). IR (neat): ν 3066, 2959, 2839, 1682, 1516, 1248, 1034, 924, 826, 716, 687. HRMS (ESI+, *m*/*z*): [M + H]^+^ calcd. for C_20_H_20_F_3_O_2_, 349.1410, found 349.1444.

##### Major Isomer

^1^H NMR: δ 2.00–2.08 (1H, m), 2.14–2.25 (1H, m), 3.83 (3H, s), 3.94 (1H, quint, *J* = 9.0 Hz), 4.18 (1H, td, *J* = 9.9, 3.9 Hz), 4.74–4.81 (2H, m), 5.44 (1H, dddd, *J* = 16.8, 10.2, 7.7, 6.8 Hz), 6.94 (2H, d, *J* = 8.4 Hz), 7.26–7.37 (2H, m), 7.45–7.53 (2H, m), 7.58–7.64 (1H, m), 8.00 (2H, d, *J* = 7.2 Hz). ^13^C NMR: δ 36.1, 44.2, 51.1 (q, *J* = 25.8 Hz), 55.2, 114.2, 118.0, 124.7 (q, *J* = 1.9 Hz), 127.1 (q, *J* = 279.9 Hz), 128.0, 128.3, 128.4, 128.7, 130.1, 130.78, 130.80, 133.0, 133.3, 137.5, 159.6 (q, *J* = 1.2 Hz), 201.5. ^19^F NMR: δ −68.24 (d, *J* = 9.3 Hz).

##### Minor Isomer

^1^H NMR: δ2.67 (2H, t, *J* = 6.9 Hz), 3.68 (3H, s), 3.86–3.89 (1H, m), 4.31 (1H, dt, *J* = 10.5, 6.3 Hz), 4.96 (1H, d, *J* = 10.2 Hz), 5.03 (1H, d, *J* = 16.8 Hz), 5.64 (1H, ddt, *J* = 16.8, 9.9, 7.2 Hz), 6.70 (2H, d, *J* = 8.7 Hz), 7.14 (2H, d, *J* = 8.7 Hz), 7.26–7.37 (3H, m), 7.68 (2H, d, *J* = 7.5 Hz). ^13^C NMR: δ 35.5 (q, *J* = 2.1 Hz), 47.0, 50.1 (q, *J* = 26.5 Hz), 55.0, 113.8, 118.3, 126.1 (q, *J* = 2.5 Hz), 130.5 (q, *J* = 289.0 Hz), 133.3, 133.5, 137.4, 159.1 (q, *J* = 1.2 Hz), 200.5. ^19^F NMR: δ −64.92 (d, *J* = 9.0 Hz).

#### 4.4.3. 4,4,4-Trifluoro-3-(4-fluorophenyl)-1-phenyl-2- (prop-2-en-1-yl)butan-1-one (**5c**)

##### Method 1

Instead of (*E*)-1,1,1-trifluoro-2,4-diphenyl-4-{(prop-2-en-1-yl)oxy}but-2-ene **3a**, 0.17 g of (*E*)-1,1,1-trifluoro-2-(4-fluorophenyl)-4-phenyl-4-{(prop-2-en-1-yl)oxy}-but-2-ene **3c** (0.50 mmol) was employed and stirring was continued for 3 h under reflux to furnish 0.15 g (0.46 mmol) of an inseparable 65:35 diastereomer mixture of the title compound was isolated in 91% yield as a colorless oil.

##### Method 2

Instead of (*E*)-4,4,4-trifluoro-1,3-diphenyl-1-(prop-2-en-1-yloxy)but-1-ene **4a**, 0.17 g of (*E*)-4,4,4-trifluoro-3-(4-fluorophenyl)-1-phenyl-1-{(prop-2-en-1-yl)oxy}-but-1-ene **4c** (0.50 mmol) was employed and stirring was continued for 5 h to furnish 0.11 g (0.33 mmol) of an inseparable 94:6 diastereomer mixture of the title compound was isolated in 65% yield as a colorless oil.

Rf = 0.37 (Hex:AcOEt = 6:1). IR (neat): ν 080, 2982, 1684, 1608, 1513, 1448, 1254, 1167, 1121, 924, 829. HRMS (FAB+, *m*/*z*): [M + H]^+^ calcd. for C_19_H_17_F_4_O, 337.1210, found 337.1202.

##### Major Isomer

1H NMR: δ 1.98–2.06 (1H, m), 2.13–2.24 (1H, m), 4.00 (1H, dq, J = 10.8, 8.6 Hz), 4.19 (1H, ddd, J = 10.5, 9.8, 4.0 Hz), 4.73–4.83 (2H, m), 5.43 (1H, dddd, J = 16.8, 10.2, 7.7, 6.6 Hz), 6.83–8.01 (9H, m). 13C NMR: δ 35.9, 44.0, 51.0 (q, *J* = 26.1 Hz), 115.9 (d, *J* = 21.1. Hz), 118.2, 126.3 (q, *J* = 278.8 Hz), 127.9, 128.3, 128.7, 131.4 (d, *J* = 8.0 Hz), 133.0, 133.4, 137.3, 162.7 (d, *J* = 247.5 Hz), 201.1. ^19^F NMR: δ −68.15 (3F, d, *J* = 9.0 Hz), −114.41~−114.30 (1F, m).

##### Minor Isomer

1H NMR: δ 2.69 (2H, t, J = 6.6 Hz), 3.87 (1H, quint, J = 9.9 Hz), 4.31 (1H, dt, J = 10.5, 6.3 Hz), 4.98 (1H, d, J = 10.2 Hz), 5.04 (1H, d, J = 17.4 Hz), 5.64 (1H, ddt, J = 16.8, 9.9, 7.2 Hz), 6.83–8.01 (9H, m). 13C NMR: δ35.4, 47.0, 50.2 (q, *J* = 27.3 Hz), 115.4 (d, *J* = 21.7 Hz), 118.6, 126.9 (q, *J* = 280.3 Hz), 128.0, 128.5, 129.9–130.0 (m), 130.7 (d, *J* = 7.5 Hz), 133.2, 137.2, 162.3 (d, *J* = 246.9 Hz), 200.2. ^19^F NMR: δ −64.77 (3F, d, *J* = 9.3 Hz), −115.00~−115.12 (1F, m).

#### 4.4.4. 1-Phenyl-2-(prop-2-en-1-yl)-3-(trifluoromethyl)pentan-1-one (**5d**)

##### Method 1

Instead of (*E*)-1,1,1-trifluoro-2,4-diphenyl-4-{(prop-2-en-1-yl)oxy}but-2-ene **3a**, 0.14 g of (*E*)-1-phenyl-1-{(prop-2-en-1-yl)oxy}-3-(trifluoro-methyl)pent-2-ene **3d** (0.50 mmol) was employed and stirring was continued for 48 h under reflux to furnish 0.12 g (0.44 mmol) of an inseparable 55:45 diastereomer mixture of the title compound was isolated in 87% yield as a colorless oil.

##### Method 2

Instead of (*E*)-4,4,4-trifluoro-1,3-diphenyl-1-(prop-2-en-1-yloxy)but-1-ene **4a**, 0.14 g of (*E*)-1-phenyl-1-{(prop-2-en-1-yl)oxy}-3-(trifluoromethyl)pent-1-ene **4d** (0.50 mmol) was employed and stirring was continued for 5 h to furnish 0.041 g (0.33 mmol) of an inseparable 57:43 diastereomer mixture of the title compound was isolated in 65% yield as a colorless oil. Because it is not possible to completely assign all the peaks to major and minor isomers, the peaks observed were described.

Rf = 0.43 (Hex:AcOEt = 10:1). ^1^H NMR: δ 0.99 (3H, q, *J* = 7.5 Hz), 1.60–1.78 (2H, m), 2.34–2.47 (1H, m), 2.50–2.71 (2H, m), 3.83–3.90 (1H, m), 4.93–5.07 (2H, m), 5.59–5.74 (1H, m), 7.46–7.51 (2H, m), 7.57–7.61 (1H, m), 7.90–7.95 (2H, m). ^13^C NMR: δ 11.9 (q, *J* = 1.3 Hz), 12.2 (q, *J* = 0.6 Hz), 17.7 (q, *J* = 1.8 Hz), 19.8 (q, *J* = 2.4 Hz), 31.4, 34.1 (q, *J* = 1.2 Hz), 43.5 (q, *J* = 1.9 Hz), 43.9 (q, *J* = 1.2 Hz), 45.4 (q, *J* = 38.5 Hz), 45.7 (q, *J* = 38.5 Hz), 117.3, 117.8, 128.0 (q, *J* = 281.6 Hz), 128.2 (q, *J* = 281.0 Hz), 128.2, 128.3, 128.7, 128.8, 133.2, 133.3, 134.4, 134.8, 136.4, 137.3, 200.5, 200.8. ^19^F NMR: δ − 66.72 (d, *J* = 9.0 Hz; **minor**), −67.96 (d, *J* = 9.0 Hz; **major**). IR (neat): ν 2975, 1683, 1597, 1448, 1253, 1170, 1141, 921, 688, 553, 523. HRMS (FAB+, *m*/*z*): [M]^+^ calcd. for C_15_H_18_F_3_O, 270.1226, found 270.1212.

#### 4.4.5. 1,5-Diphenyl-2-(prop-2-en-1-yl)-3-(trifluoromethyl)pentan-1-one (**5e**)

##### Method 1

Instead of (*E*)-1,1,1-trifluoro-2,4-diphenyl-4-{(prop-2-en-1-yl)oxy}but-2-ene **3a**, 0.17 g of (*E*)-1,5-diphenyl-1-{(prop-2-en-1-yl)oxy}-3-(trifluoromethyl)pent-2-ene **3e** (0.50 mmol) was employed and stirring was continued for 18 h under reflux to furnish 0.14 g (0.50 mmol) of an inseparable 55:45 diastereomer mixture of the title compound was isolated in 84% yield as a colorless oil. Because it is not possible to completely assign all the peaks to major and minor isomers, the peaks observed were described.

Rf = 0.51 (Hex:AcOEt = 10:1). ^1^H NMR: δ 1.87–1.98 (2H, m), 2.30–2.42 (1H, m), 2.53–2.83 (4H, m), 3.81–3.91 (1H, m), 4.93–5.07 (2H, m), 5.55–5.73 (1H, m), 7.07–7.29 (5H, m), 7.42–7.46 (2H, m), 7.47–7.57 (1H, m), 7.82–7.87 (2H, m). ^13^C NMR: δ 26.1 (q, *J* = 1.8 Hz), 28.2 (q, *J* = 1.9 Hz), 30.9, 33.4, 33.7, 34.1, 43.0 (q, *J* = 24.8 Hz), 43.9 (q, *J* = 24.8 Hz), 43.8 (q, *J* = 1.8 Hz), 44.0 (q, *J* = 1.2 Hz), 117.3, 117.9, 126.1, 126.2, 128.0 (q, *J* = 281.0 Hz), 128.1 (q, *J* = 281.0 Hz), 128.16, 128.21, 128.3, 128.4 (2C), 128.5, 128.70, 128.73, 133.19, 133.23, 134.3, 134.8, 136.1, 137.2, 140.5, 140.7, 200.0, 200.6. ^19^F NMR: δ −66.99 (d, *J* = 9.0 Hz; **minor**), −68.04 (d, *J* = 9.0 Hz; **major**). IR (neat): ν 3064, 3028, 2953, 1685, 1448, 1254, 1155, 1117, 1001, 921, 700. HRMS (ESI+, *m*/*z*): [M + H]^+^ calcd. for C_21_H_22_F_3_O, 347.1617, found 347.1618.

#### 4.4.6. 1-(4-Methoxyphenyl)-5-phenyl-2-(prop-2-en-1-yl)-3-(trifluoromethyl)pentan-1-one (**5f**)

##### Method 1

Instead of (*E*)-1,1,1-trifluoro-2,4-diphenyl-4-{(prop-2-en-1-yl)oxy}but-2-ene **3a**, 0.19 g of (*E*)-1-(4-methoxyphenyl)-5-phenyl-1-{(prop-2-en-1-yl)oxy}-3-(trifluoromethyl)pent-2-ene **3f** (0.50 mmol) was employed and stirring was continued for 18 h under reflux to furnish 0.11 g (0.28 mmol) of an inseparable 50:50 diastereomer mixture of the title compound was isolated in 56% yield as a colorless oil. Because it is not possible to completely assign all the peaks to major and minor isomers, the peaks observed were described.

Rf = 0.34 (Hex:AcOEt = 10:1). ^1^H NMR: δ 1.85–1.99 (2H, m), 2.30–2.40 (1H, m), 2.54–2.78(4H, m), 3.77–3.87(1H, m), 3.865 (3H, s), 3.869 (3H, s), 4.92–5.07 (2H, m), 5.54–5.72 (1H, m), 6.89–6.95 (2H, m), 7.08–7.29 (5H, m), 7.83–7.90 (2H, m). ^13^C NMR: δ 26.2 (q, *J* = 1.8 Hz), 28.3 (q, *J* = 2.5 Hz), 31.0, 33.5 (q, *J* = 1.3 Hz), 33.8 (q, *J* = 1.3 Hz), 34.6 (q, *J* = 1.3 Hz), 43.3 (q, *J* = 24.8 Hz), 43.3 (q, *J* = 1.9 Hz), 43.5 (q, *J* = 1.9 Hz), 44.0 (q,*J* = 24.8 Hz), 55.41, 55.43, 113.89, 113.93, 117.1, 117.8, 126.07, 126.11, 128.1 (q, *J* = 281.6 Hz), 128.2 (q, *J* = 280.4 Hz), 128.3, 128.38, 128.40, 128.5, 129.0, 130.3, 130.6 (2C), 134.5, 135.1, 140.7, 140.9, 163.7 (2C), 198.4, 199.0. ^19^F NMR: δ −66.96 (d, *J* = 11.3 Hz), −68.10 (d, *J* = 9.3 Hz). IR (neat): ν 3064, 2938, 1675, 1601, 1510, 1255, 1172, 1116, 1031, 843, 700. HRMS (FAB+, *m*/*z*): [M]^+^ calcd. for C_22_H_23_F_3_O_2_, 376.1645, found 376.1692.

#### 4.4.7. 1-(4-Bromophenyl)-5-phenyl-2-(prop-2-en-1-yl)-3-(trifluoromethyl)pentan-1-one (**5g**)

##### Method 1

Instead of (*E*)-1,1,1-trifluoro-2,4-diphenyl-4-{(prop-2-en-1-yl)oxy}but-2-ene **3a**, 0.21 g of (*E*)-1-(4-bromophenyl)-5-phenyl-1-{(prop-2-en-1-yl)oxy}-3-(trifluoromethyl)pent-2-ene **3g** (0.50 mmol) was employed and stirring was continued for 18 h under reflux to furnish 0.18 g (0.42 mmol) of an inseparable 55:45 diastereomer mixture of the title compound was isolated in 85% yield as a colorless oil. Because it is not possible to analyze these peaks completely, the peaks observed were described.

Rf = 0.34 (Hex:AcOEt = 20:1). ^1^H NMR: δ 1.90–2.00 (2H, m), 2.26–2.40 (1H, m), 2.54–2.84 (4H, m), 3.71–3.82 (1H, m), 4.93–5.07 (2H, m), 5.52–5.69 (1H, m), 7.06–7.31 (5H, m), 7.56–7.63 (2H, m), 7.66–7.71 (2H, m). ^13^C NMR: δ 26.1 (q, *J* = 1.9 Hz), 27.8 (q, *J* = 1.9 Hz), 31.0, 33.2, 33.7, 33.9, 42.8 (q, *J* = 24.1 Hz), 43.759 (q, *J* = 1.9 Hz), 43.763 (q, *J* = 24.8 Hz), 43.9 (q, *J* = 1.9 Hz), 117.6, 118.1, 126.1, 126.2, 127.9 (q, *J* = 281.0 Hz), 128.0 (q, *J* = 284.1 Hz), 128.27, 128.31, 128.4, 128.46, 128.48, 128.53, 129.6, 129.7, 131.97, 132.0, 134.2, 134.5, 134.8, 135.9, 140.4, 140.6, 199.0, 199.6. ^19^F NMR: δ −67.15 (d, *J* = 11.3 Hz; **minor**), −68.03 (d, *J* = 9.0 Hz; **major**). IR (neat): ν 3064, 3028, 2949, 1685, 1585, 1397, 1254, 1158, 923, 747, 700. HRMS (FAB+, *m*/*z*): [M + H]^+^ calcd. for C_21_H_21_BrF_3_O, 425.0722, found 425.0757.

## Data Availability

Not applicable.

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
