# Peer review of "Base-Mediated Claisen Rearrangement of CF3-Containing Bisallyl Ethers"

_molecules, 2021, doi:10.3390/molecules26144365_

Round 1

Reviewer 1 Report

Molecules-1287197

Title: Base-Mediated Claisen Rearrangement of CF3-Containing Bisallyl Ethers

Authors: Yoko Hamada, Rio Matsunaga, Tomoko Kawasaki-Takasuka, Takashi Yamazaki

The authors reported several Claisen-rearrangements of CF3-containing bisallyl ethers with different bases, this research is well described. The diastereomeric ratios were changed presents of Pd2+-ions at room temperature. The authors gave a good mechanism to support their results using previous studies as well. I think that this MS should be published in Molecules after a minor revision.

My comments:

My general impression is that some paragraphs are missing due to bad formatting or using the Molecules template (I highlighted in MS.). Before the resubmission, please check the whole paper again.

1./ The Scheme 1 can be reduced in size and insert Page 1, Line 27. This will help the readers to better understand the following sections.

2./ Table 1 should be inserted to Line 64. In Table 1, the temperature column can be deleted.

3./ Some abbreviations are missing from the manuscript (e.g. DBU, HMPA, LDA, DCM etc) please define these.

4./ PdCl2·(PhCN)2 is not correct please use [PdCl2(PhCN)2] or [PdCl2(NCC6H5)2]. I read that you used this salt as catalyst in Ireland-Claisen rearrangement. My question is that why you did not use only PdCl2 or Pd-acetate as the catalyst.

5./ Scheme 2 and the header of Table 3 are the same, please eliminate one of them.

6./ The manuscript mentions in several places that 19F NMR spectroscopy was used to determine isomer rations and the conversions or yields. Please give these 19F-NMR spectra for at least compounds 5a-g in the SI, in addition, these are new compounds.

I see that the 13C-NMR spectra are calibrated to the chemical shift of CDCl3 (77.00 ppm). Please correct it in the MS.

7./ In the "Author Contributions" part, please use the standard statements of MDPI, see in the user guide.

8./ DCM and CHCl3 are undesirable solvents according to the Pfizer solvent selection guide. What do you think these reactions are working with suitable yields in greener solvents like alcohols, ethyl acetate or dimethyl-carbonate?

Overall, I think it is an interesting work in the field of Fluorous Chemistry following previous research of the group. After the revision, I feel it will fit in Molecules, Organofluorine Chemistry and Beyond special issue.

Author Response

Thank you very much for your valuable comments and please find our reply to them.

Answer to the comment 1.  We have change Scheme 1 which moved to page 1.

Answer to the comment 2.  Table 1 moved to page 3.  It is most appropriate to put this Table to Line 64, while it made this Table divided into two.  After checking other articles published from Molecules, it seems that it is not a rule to put Tables, Schemes, and Figures right after appearance of these words in the text, and thus, we used this placement after some trial.

Answer to the comment 3.  We added “full names” for abbreviated ones.

Answer to the comment 4.  For description of the catalyst, we followed to your suggestion.  In the case of the catalyst, to the best of my knowledge, the first report using a palladium catalyst was published in 1986 when [PdCl2·(PhCN)2] was employed and Pd(OAc)2was known to follow a different mechanistic route.  Moreover, as written in the text, because [PdCl2·(PhCN)2] worked quite nicely in our previous case using similar rearrangement systems, we did not use other Pd(II) catalysts although they would give some interesting outcome.

Answer to the comment 5.  Actually, this occurred when the pdf version was made possibly by the editorial staff (because no such problems were found for the one we have sent to them).  So, there is no problem in the text made by MS word.

Answer to the comment 6.  As long as we concern, SI are used for confirmation of purities of isolated compounds and usually, only 1H and 13C NMR are included.  Except for the compounds after rearrangement, only one CF3 peaks are found in all compounds and we actually don’t understand the importance of 19F NMR charts to add to SI.  So, we did not change SI at all.  It is really appreciated if we are really required to add them, it is really appreciated if you kindly inform to us.  In the case of 13C NMR, we corrected the “4.1 General Information” following to your suggestion.

Answer to the comment 7.  We corrected them by following to your suggestion.

Answer to the comment 8.  Usage of greener solvents are of course preferable, but in our case, for example, the O-allylation process requires appropriate base like NaOH aq. and thus, ethyl acetate and dimethyl carbonate are not adequate due to their possible hydrolysis.  Moreover, since the reaction of the OH function of our compounds is required in this process, ethanol is apparently not suitable.  In the case of our last examples using Lewis acidic additives, these greener solvents should easily construct complexes with them.  So, we really understand and agree what this reviewer mentioned, but it is just a case by case and we will try such solvents in the future work if they are appropriate

Reviewer 2 Report

The paper of Yamazaki and coworker presents a base-mediated proton migration reactions of CF3-containing allylic alcohols to afford the corresponding a,b-(un)saturated ketones which was submitted to Claisen rearrangement.

In continuation of their important previous work reporting proton transfer studies from both propargylic and allylic alcohols, here the approach describes by the authors allows the formation of type-5 compounds by way of Claisen rearrangement.

The authors accurately explain the advantages of this approach nevertheless, about readability of the work some remarks have arisen:

- End of Page 1 : ... to the corresponding allyl… ?

- Caption of Scheme 1 isn’t appropriate

- Table 1 presents optimization of the reaction conditions for the O-allylation of compound 1a which is not yet described (nature of R1 and R2).

Moreover this table should be simplified

- Page 3, section 2.1. Preparation of bisallyl ethers 3 :

… Final examination on the… ?

And the footnote of the Table 2 doesn’t correspond.

- Section 2.2. Preparation of allyl vinyl ethers 4 and one-pot isomerization-Claisen rearrangement :

Line 105 ; Could the authors explain : « Stereochemistry of 5a was presumed to be Z » ?

- Section 2.3. Improvement of the diastereoselectivity of the Claisen rearrangement products 5.

Scheme 2 «Effect of DBU for the epimerization of the rearranged product 5a» should appear in this section or should be delete regarding Table 4.

Please homogenise in Tables : yields determined by 19F NMR vs yields after isolation in brackets  vs diastereomer ratios in parentheses.

In conclusion, the manuscript of Yamazaki et al., could be accepted but after major text corrections.

Others remarks :

- Abstract : CF3-containing

- Page 1, L36, bisallylic ethers 3 (not 2)

- Table 1,         entry 6 : BnEt3NCl

                        Footnote 5 : 3.0 equiv of HMPA was added.vinyl ethers 4 ?

Author Response

Thank you very much for your valuable comments and please find our reply to them.

Answer to the comment from “- End of Page 1 :” to “the footnote of the Table 2”, “- Section 2.3”, and the last two in the “Other remarks”.  Actually, this occurred when the pdf version was made possibly by the editorial staff (because no such problems were found for the one we have sent to them).  So, there are no problems in the text made by MS word.

Answer to the comment of - Section 2.2.  We are sorry about this mistake, and it is not 5a but 4a we should describe here.

Answer to the comment of “Please homogenise in Tables”.  We have change them following to your suggestion.

Answer to the comment of the first two in the “Other remarks”.  We have change them following to your suggestion.

Round 2

Reviewer 2 Report

In this revised version the autors have taken into account a majority of reviewers comments and the readability of this article is now acceptable.

In conclusion the manuscript of Yamazaki and coworkers is recommended for publication in Molecules

Remark: Table 4: Claisen rearrangement of 4 mediated by additives